# SePA: A Search-enhanced Predictive Agent for Personalized Health Coaching

Melik Ozolcer and Sang Won Bae
*Department of Systems Engineering*
*Stevens Institute of Technology*
Hoboken, NJ, USA
{mozolcer, sbae4}@stevens.edu

*Abstract*—This paper introduces SePA (Search-enhanced Predictive AI Agent), a novel LLM health coaching system that integrates personalized machine learning and retrieval-augmented generation to deliver adaptive, evidence-based guidance. SePA combines: (1) Individualized models predicting daily stress, soreness, and injury risk from wearable sensor data (28 users, 1260 data points); and (2) A retrieval module that grounds LLM-generated feedback in expert-vetted web content to ensure contextual relevance and reliability. Our predictive models, evaluated with rolling-origin cross-validation and group 4-fold cross-validation show that personalized models outperform generalized baselines. In a pilot expert study (n=4), SePA's retrieval-based advice was preferred over a non-retrieval baseline, yielding meaningful practical effect (Cliff's $\delta$=0.3, p=0.05). We also quantify latency performance trade-offs between response quality and speed, offering a transparent blueprint for next-generation, trustworthy personal health informatics systems.

*Index Terms*—Large Language Models, Personalized Health, Wearable Sensors, Predictive Modeling

## I. INTRODUCTION

The proliferation of wearable sensors has ushered in an era of unprecedented personal health data, offering the potential to move from reactive healthcare to proactive wellness management. Consumer devices from Fitbit, Garmin, and Apple now continuously stream physiological and behavioral data, yet a critical gap persists: the raw data or simple trend visualizations provided by most applications often fail to translate into proactive, personalized, and trustworthy guidance [1]. Users are left with metrics, but little understanding of what to do next to mitigate future risks of stress, soreness, or injury. Large Language Models (LLMs) have emerged as a promising technology to bridge this gap. Early systems like PhysioLLM demonstrated that integrating wearable data with LLM-driven interfaces could yield richer, more personalized insights for domains like sleep hygiene and recovery [2]. Subsequent research, including PH-LLM [3] and Health-LLM [4], advanced this paradigm by leveraging sophisticated models trained on multimodal health records to provide patient-centric explanations. While powerful, these systems have largely remained retrospective, excelling at explaining historical data rather than forecasting near-future wellness states.

Recent work such as PHIA [5] highlights the potential of LLMs to dynamically interact with personal data and external knowledge sources. While promising, it lacks open-source accessibility, personalized retrieval based on biometric context, and verified sourcing needed for health-related guidance.

This paper introduces SePA (Search-enhanced Predictive Agent), an LLM health agent designed with the principles of transparency and evidence-based coaching, to overcome these limitations. Our system integrates proactive risk prediction directly into a trustworthy, context-aware conversational loop. We move beyond reactive analysis by forecasting daily, subjective states of stress, muscle soreness, and injury risk from wearable data. These predictions then serve as dynamic context for a web-retrieval pipeline that grounds its advice in a whitelist of trusted sources, ensuring all claims are verifiable and relevant to the user's current and predicted state.

Our primary contributions are as follows:

- Development and validation of a two-tiered predictive modeling strategy for daily stress, soreness, and injury risk. This strategy balances immediate utility of a generalized model (validated with group 4-fold cross-validation) with the superior accuracy of personalized neural models, which we evaluate using rolling-origin cross-validation.
- A context-aware and trusted web-retrieval pipeline that dynamically rewrites search queries using daily ML-driven risk predictions. This ensures the retrieved content is both contextually relevant to the user's physiological state and sourced from expert-vetted domains.
- A transparent architectural blueprint and performance analysis, including detailed system design and documentation.
- Preliminary expert validation through a blind evaluation with four domain experts, demonstrating that our web-search retrieval-augmented coaching agent provides recommendations of meaningfully higher quality, relevance, and helpfulness compared to a non-retrieval baseline.

Our web-retrieval pipeline implementation is available at https://github.com/stevenshci/sepa-web-search.

By integrating proactive forecasting with verifiable, context-aware retrieval, SePA presents a significant step toward the next generation of digital health agents that are not only intelligent but also predictive, transparent, and trustworthy.

## II. RELATED WORKS

Our research builds upon three distinct but converging lines of work: predictive modeling from wearable data, the

development of LLM-powered health agents, and the use of web-retrieval for trustworthy guidance.

### A. Predicting Stress, Soreness, Injury Risk via Wearable Data

The last five years have witnessed remarkable advances in automated wellness prediction using wearable sensors. Early research established strong correlations between physiological signals like heart rate variability (HRV) and sleep disruption with perceived stress [6]. More recent work has evolved from retrospective analysis to forecasting, with some models achieving F1 scores above 0.80 for next-day stress prediction by integrating heart rate, accelerometry, and contextual data [7].

Predicting muscle soreness, particularly delayed onset muscle soreness (DOMS), remains a challenge due to its subjective nature. However, wearable-derived features such as training load, high-intensity movement counts, and heart rate zone exposures have been shown to significantly anticipate next-day soreness [8]. Similarly, the prediction of injury risk has been a major focus, often employing tree-based methods or logistic regression [9], [10]. Many of these models grapple with severe class imbalance from infrequent injury events [11], and while some report high performance, broad injury definitions can limit their practical relevance in specific athletic contexts [12]. A recurring theme in this domain is the high inter-individual variability, suggesting that personalized models often outperform generalized ones. Our work builds on these foundations by developing personalized models specifically for proactive, daily forecasting to power a downstream agent.

### B. LLM Health Agents Leveraging Wearable Data

The intersection of wearables and LLMs has catalyzed a new wave of personal health technologies. PhysioLLM was a pioneering system that demonstrated how combining statistical analysis of Fitbit data with an LLM summarizer could produce more actionable, user-centered coaching than traditional dashboards [2]. This line of work was extended by more sophisticated systems like PH-LLM [3] and Health-LLM [4], which utilized large-scale, transformer-based models to provide patient-centric explanations and recommendations from multimodal health records. Other approaches, such as GPTCoach [13], have focused on leveraging LLMs for motivational interviewing and goal setting.

Despite their sophistication, these systems are primarily reactive. Their main function is to interpret and explain historical trends, with limited support for actionable prediction and prevention. They excel at answering *what happened?* but are less equipped to answer *what is my risk tomorrow, and what should I do about it?*. SePA is designed specifically to be proactive, using daily risk forecasts as the primary driver for its conversational guidance.

### C. Live Web-Retrieval for Personal-Health LLM Agents

To ensure advice is both current and factually accurate, researchers enhance the LLM context with retrieval-augmented generation (RAG). In medicine, RAG has proven essential for reducing factual errors and hallucinations, with studies showing it can cut unsupported claims in clinical guidance from 23% down to just 4% [14]. A recent meta-analysis confirmed this, finding a pooled 1.35x performance lift when RAG is added to a health-oriented LLM [15]. Systems have integrated RAG for sleep-hygiene coaching [16], diabetes meal planning [17], and other wellness domains. However, these often rely on static corpora or generic web searches without strong source filtering, limiting trustworthiness. Wu et al. showed that GPT-4 with unrestricted web search still produced unsupported statements 30% of the time in medical responses [18]. This underscores that the quality of retrieved content and its integration are crucial for trustworthy health guidance.

PHIA [5] represents the state-of-the-art in the context of LLM-based health coaching, introducing a ReAct-style agent that combines data analysis with Google Search. While influential, the system has critical limitations for real-world deployment: (1) it lacks contextual retrieval, as searches are not personalized using real-time biometrics or risk predictions; (2) it retrieves from the unrestricted public web without verified sources or trust mechanisms; and (3) it is not open-source, with key components of its retrieval pipeline and prompts undocumented, limiting reproducibility.

Our work directly addresses these gaps by introducing a privacy-preserving agent architecture designed for trust and transparency. Its novel web-retrieval pipeline, which we commit to releasing as open-source, is both context-aware, using ML predictions to inform retrieval, and trust-filtered, using a rigorously curated domain whitelist and strict citation requirements.

## III. System Architecture & Method

SePA is an end-to-end system designed to provide proactive, personalized, and trustworthy health guidance. Its architecture integrates three core components: (1) an asynchronous data processing and prediction pipeline, (2) an agentic conversational layer, and (3) a novel trusted, context-aware web-retrieval pipeline. A high-level overview of the system is presented in Figure 1.

### A. Overall Architecture and Data Flow

The system is implemented as a web application with a Python Flask backend. User management and persistence are handled via a PostgreSQL database.

1) Data Ingestion: Users upload their Apple Health data as a ZIP archive. This triggers an asynchronous process_uploaded_data_task managed by a Celery worker queue with a Redis broker. This architecture allows for concurrent processing of multiple uploads without impacting conversational latency.

2) Preprocessing & Feature Engineering: The worker unpacks the raw data, cleans it, and engineers a daily feature matrix. Key variables include sleep stages, wearable tracked vitals, and activity summaries (steps, calories, etc.) (See Section III-B for more details). The original

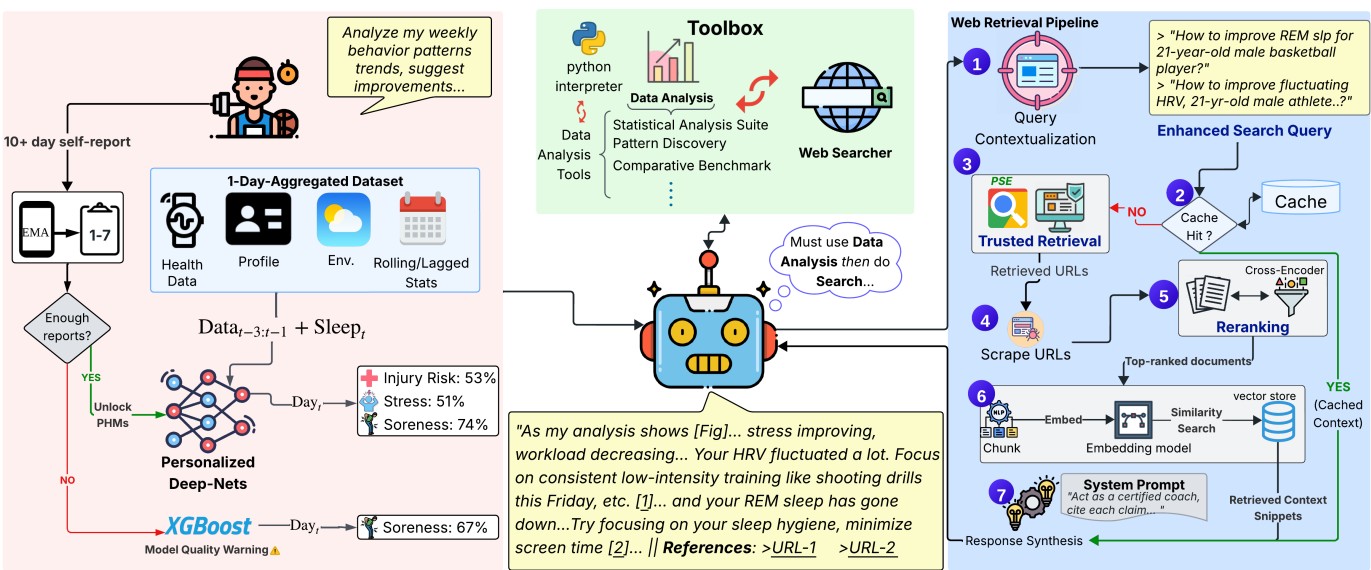

Fig. 1: SePA system architecture overview. User uploads their Apple Health data export to our website, which is then transformed into a 1-Day aggregated tabular data format after feature extraction. If the user provides 10+ days of labels for their subjective feelings of injury risk, soreness, stress, our higher performing Personalized Health Models (PHMs) are unlocked. For cold-start scenario, the user has access to a lower-performing, exploratory XGBoost-Soreness model. For incoming user query, SePA analyzes which tool(s) is needed. For web-retrieval, pipeline flow and processing steps are depicted on the right (Reranking is done for pre-filtering purposes). The example response illustrates the final output, demonstrating how the system synthesizes personalized data with cited web content to provide actionable, evidence based guidance.

raw data is permanently deleted after this step to enforce privacy, with only the derived feature set being stored.

3) Conversational Loop: The user interacts with the LLM agent-powered by any OpenAI API compatible model with tool call support- via the web interface, including open-source LLMs served from Groq API. whose privacy guarantees are detailed in Section III-D. The agent maintains conversational history and has access to a suite of tools, which it can autonomously invoke to answer user queries.

### B. Proactive Health Predictions

A core contribution of our system is the move from post-hoc analysis of health data via a chat interface, to proactive forecasting enabling preventive health insights. Each morning after wake-up, we predict self-reported stress, soreness, and injury risk scores for the current day ahead, enabling athletes to receive actionable health insights before starting their day. After recording sleep data each morning, our models forecast the day's average self-reported health scores (spanning morning, afternoon, and evening). The predictions use the past 72 hours of wearable and environmental data, the just-completed night's sleep-related metrics (including vitals like HRV, $SpO_2$, $VO_2$ max, Resp Rate.), with the target being the average of three same-day self-reports (1-7 scale) collected at morning, afternoon, and evening. We gathered continuous wearable data from Fitbit devices alongside self-reported health ratings from student-athletes at our institution. The day-aggregated dataset includes all wearable captured data,

environmental factors (weather), and athlete demographics (age, weight, height, sport), processed using statistical methods ranging from basic (mean, std, skewness) to advanced (DFA, entropy, second-order statistics [19]) for heart rate and sleep.

Our analysis revealed that purely generalized models struggle with the high inter-individual variability of these subjective health states, achieving more limited predictive power. We therefore designed a practical, two-tiered deployment strategy that transitions from general to personalized models as user data accumulates.

*1) Tier 1: Generalized Model for New Users (Cold-Start):* For a new user with no historical labels, the system deploys a generalized XGBoost (G) model trained on pooled data from our entire athlete cohort. While this model performs poorly for stress and injury risk (negative $R^2$ in group cross-validation, see Figure 4), it achieves a modest but positive predictive signal for soreness ($R^2 \approx 0.15$). This provides immediate, preliminary guidance to new users, encouraging engagement without an initial labeling burden.

*2) Tier 2: Personalized Model for Engaged Users:* Once a user provides sufficient daily labels ($d > 15$ days), the system switches to our Personalized Health Models (PHMs). The detailed methodology, dataset, and validation for the injury risk component of this model are presented in [20]. Our proposed neural network architecture (Fig. 2) features participant-specific embeddings, learned numerical vectors that allow the model to learn each individual's unique physiological baseline and response patterns. The embedding concatenated before feature extraction captures individual physiological baselines,

while the second concatenation enables person-specific prediction scaling. To address overfitting given our dataset of 28 participants (1,260 participant-days), we employed multicollinearity removal and F-test feature selection, reducing features from 200+ to 60-70 per task. As shown in Figure 3, this personalized approach improves performance, achieving $R^2 > 0.50$ for stress, $R^2 > 0.40$ for injury risk, and $R^2 \approx 0.28$ for soreness.

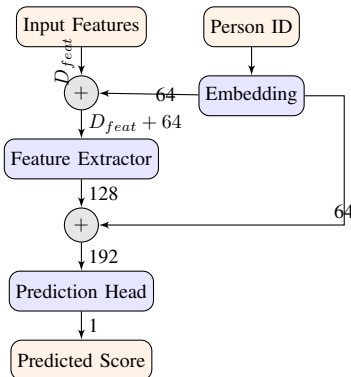

Fig. 2: PHM architecture with double concatenation of 64-dim person embeddings. Feature extractor (input+64→256→128) and prediction head (192→64→1) with ReLU activations and dropout (p=0.5). L2 regularization ($\lambda$=1e-5) for training.

### C. Trusted, Context-Aware Web-Retrieval Pipeline

When a user asks an advice-seeking question, the agent invokes our novel Web-Retrieval pipeline.

1) **Query Contextualization:** The pipeline augments each user query with personal context including demographics (age, sex, sport), recent data-driven insights, and current ML risk predictions (stress, soreness, injury). This transforms a generic question (e.g., "How can I reduce soreness?") into a privacy-preserving search prompt (e.g., "Strategies to reduce soreness for a 21-year-old basketball player with high soreness (74%) and elevated RHR").

2) **Cache Check:** Before searching, a multi-layer cache (in-memory, disk, semantic) is queried with the augmented prompt. Cache hits bypass steps 3–6 and go straight to synthesis.

3) **Trusted Retrieval:** On a cache miss, the query is sent to Google Programmable Search Engine restricted to a curated whitelist (35 trusted domains: professional societies, major medical centers, PubMed). A separate branch handles video requests.

4) **Scraping & Cleaning:** Returned URLs are fetched asynchronously (`aiohttp`); boilerplate is removed and main text extracted.

5) **Document-Level Reranking:** Pages are scored against the augmented query with a cross-encoder to filter before embedding.

6) **Semantic Similarity Search:** Top-ranked documents are (a) split into 800-character chunks, (b) embedded via an embedding model, and (c) indexed in a `FAISS` vector store; the enhanced query then similarity-searches this index to retrieve the most relevant snippets for the final response.

7) **Response Synthesis:** Retrieved snippets plus the system prompt ("act as a certified sports-medicine coach; cite every factual claim") guide the LLM to produce an answer with inline citations and a source list.

### D. Privacy and Open Implementation

Our architecture ensures privacy through three layers: (1) Raw user health data is ephemeral and deleted post-processing. (2) No personally identifiable information is ever sent to external APIs (both LLM and Google PSE APIs); only the anonymized, rewritten search queries are transmitted (no name information being passed to the context). (3) We give users the option to use no-retention LLM endpoints, such as the Groq API, which process queries for immediate inference and then discard all data[1]. The web-retrieval pipeline implementation, full domain whitelist, and prompt templates are publicly available to ensure reproducibility.

## IV. EVALUATION

We conducted a two-part evaluation to validate the core components of SePA. First, we assessed the performance and necessity of our predictive modeling strategy. Second, we conducted an expert-driven study to measure the quality of the guidance generated by our web-retrieval pipeline.

### A. Predictive Model Performance

As detailed in Section III-B, our analysis focused on the performance of personalized versus generalized models. The results, summarized in Figure 3 and Figure 4, confirm our central hypothesis.

The within-participant rolling-origin cross-validation evaluation (Figure 3) shows that personalized models, particularly our PHM model, consistently and significantly outperform all generalized baselines across all three tasks (stress, soreness, and injury risk) once a sufficient amount of historical data is available. The PHM model reaches a robust coefficient of determination ($R^2$) of over 0.40 injury risk, and over 0.50 for stress predictions, demonstrating sufficiently strong predictive power.

Conversely, the group 4-fold cross-validation (Figure 4), which simulates model performance on completely unseen users, highlights the severe limitations of a generalized approach. For stress and injury risk, the best-performing global models yielded negative $R^2$ values, indicating their predictions were worse than simply using the dataset's mean. Only for soreness did the XGBoost (G) model show a modest positive performance ($R^2 \approx 0.15$). These quantitative results provide strong evidence for our two-tiered deployment strategy, which balances the need for immediate utility in a cold-start scenario with higher accuracy of a personalized approach for engaged users.

---

[1] www.groq.com/privacy-policy/

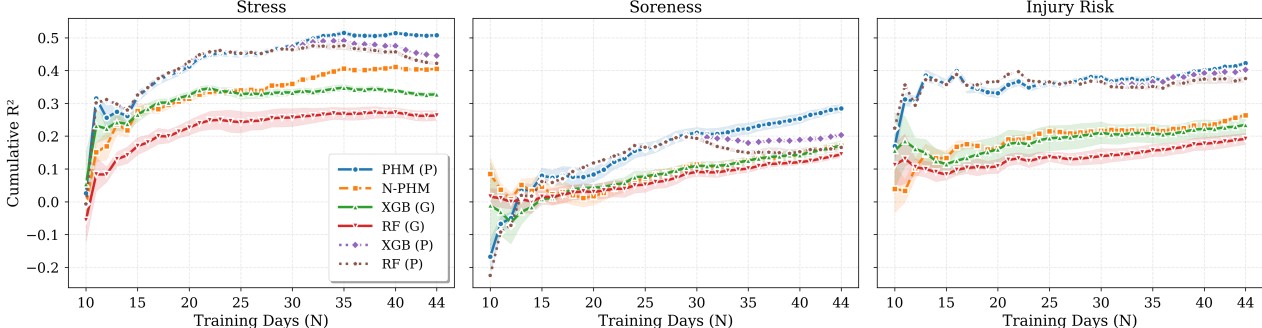

Fig. 3: Cumulative model performance comparison across stress, soreness, and injury risk prediction tasks. Models were evaluated using rolling-origin validation with first N training days and tested on day N+1. Cumulative $R^2$ values incorporate all test predictions from day 10 to N. PHM (P): our proposed personalized model with participant embeddings; other baselines include non-personalized and traditional ML approaches. Results averaged over 5 runs (Std. shown in shaded areas).

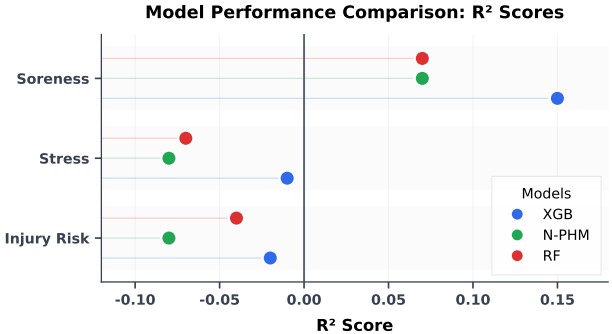

Fig. 4: Model performance ($R^2$) comparison using group 4-fold cross-validation (group-out CV, unseen participants). PHM without personal embeddings (N-PHM), XGBoost (General) model for soreness has a positive $R^2$, RF (General) models. Noting that only soreness models achieve positive $R^2$, with XGBoost achieving the highest value at 0.15.

### B. Web-Retrieval-Augmented Coaching Quality

To assess whether our context-aware web-retrieval pipeline improves the quality of health coaching, we conducted a blind evaluation with four domain experts: a Professor of Psychology with expertise in wellness coaching, head coaches of field-hockey and soccer teams at our institution, and the head strength and conditioning coach in our institution.

*1) Methodology:* The experts were presented with answers to ten representative advice-seeking coaching queries. The queries were prepared by drawing from existing literature [3], recorded interactions from our pilot user tests, and with the input from a Professor of Psychiatry. The queries and the inter-rater reliability (Kendall's W) scores are presented in Table I. The study design (4 experts, 2 items) constrains the Kendall's W statistic in Table I to discrete values of 1.0 (unanimity), 0.25 (3-1 majority), or 0.0 (2-2 split) which limits the metric's resolution in this context.

For each query, two responses were shown generated by:

- SePA-no-web: The LLM agent using only the user's

TABLE I: Questions and corresponding inter-rater reliability scores (Kendall's W).

| Id | Question | W |
|---|---|---|
| Q1 | How can I effectively balance my acute and chronic workload ratio (ACWR) to minimize injury risk? | 1.00 |
| Q2 | What nutritional strategies might help improve VO2? | 0.25 |
| Q3 | Could you give me a suggested workout about an hour long that hits legs, based on my current injury risk level? | 1.00 |
| Q4 | What strategies can help me optimize my REM sleep? | 0.25 |
| Q5 | How should my training volume adjust based on changes in my resting heart rate? | 1.00 |
| Q6 | I went out partying last night and got back to my apartment hammered at 4 AM. What advice would you give to help me prepare for tomorrow's game? | 0.00 |
| Q7 | I have been dealing with a Grade 2 calf strain past 2 weeks... What should I do? | 0.25 |
| Q8 | What are some recommendations or insights on how I can optimize my exercise routine and overall wellness? | 0.00 |
| Q9 | How do I reduce stress? | 1.00 |
| Q10 | How can I improve my muscle recovery? | 0.25 |

personal data, without web retrieval.
- SePA-web: Our full system, using both personal data and the trusted web-retrieval pipeline.

The experts, blind to the condition, ranked the two responses for each query on overall quality, considering accuracy, relevance, helpfulness, and completeness (1 = better, 2 = worse). All responses were generated using GPT-4o, with same exact system prompts and data availability, the only difference being the LLM not having access to the web searcher tool.

*2) Results:* As shown in Table II, the web-retrieval augmented system (SePA-web) was strongly preferred by the experts. SePA-web achieved a superior mean rank of 1.35 compared to 1.65 for the SePA-no-web system and received 26 out of 40 first-place votes.

A one-tailed Wilcoxon signed-rank test was used to evaluate the quality of recommendations. The result provided evidence against the null hypothesis at the margin of conventional

TABLE II: Comparison of Expert Rankings for Coaching Responses.

| Metric | SePA-web | SePA-no-web |
|---|---|---|
| Mean rank (↓ better) | 1.35 | 1.65 |
| # of first-place votes | 26 | 14 |

TABLE III: Summary of Qualitative Feedback from Domain Experts.

| Expert | Key Feedback Summary |
|---|---|
| Professor of Psychology (Wellness Coaching Expert) | • Praised the combination of personalized data with credible web content.
• Suggested simplifying language to match user health-literacy levels.
*"Health literacy should be considered... boil down the language."* |
| Head Strength & Conditioning Coach | • Valued detailed, cited answers for building trust.
• Stressed the agent's need to ask clarifying questions before advising on complex issues like injury.
*"Asking the right question...is the most crucial aspect."* |
| Head Field-Hockey Coach | • Highlighted the value of privacy, allowing athletes to ask "embarrassing or silly" questions without judgment.
• Noted the system's potential to support off-season self-coaching.
*"They can ask any silly question...this system can give them that resource."* |
| Head Soccer Coach | • Appreciated the utility for athletes without direct staff access, especially during the off-season.
• Emphasized the need to position the tool to support, not replace, professional staff.
*"...wouldn't like athletes disagreeing with coach saying 'my AI told me xxx'."* |

statistical significance (W = 287, p = 0.05). While this result is at the threshold of significance, it is supported by a Cliff's $\delta$ of 0.30, indicating a medium practical effect. This suggests that the improvement offered by the web-retrieval pipeline is not only statistically detectable but also meaningful in practice.

*3) Qualitative Feedback:* : Following the ranking task, experts engaged in free-form interaction with the live SePA-web system. Their qualitative feedback reinforced the quantitative findings. Experts praised the system's ability to provide *up-to-date guidelines tied to the athlete's recent workload* and appreciated the *privacy to ask embarrassing questions* without judgment. The head strength and conditioning coach noted that providing detailed, cited answers was crucial, stating that the agent should *help them know where the answer is coming from instead of just spewing the response*. This feedback directly validates our design choices of integrating context-aware retrieval and enforcing strict, verifiable citations. A summary of the qualitative feedback is presented in Table III.

## V. DISCUSSION

We offer a practical blueprint for integrating proactive, personalized prediction with trusted, context-aware retrieval in digital health coaching. By anticipating wellness risks and grounding guidance in vetted sources, SePA helps move beyond reactive *what happened?* analyses toward actionable *what might happen, and what should I do?* support.

### A. Findings and Implications

Our evaluation yielded two principal findings. First, we identified that one-size-fits-all approach is insufficient for predicting subjective health states like stress, and injury risk. Our personalized models, evaluated with rolling-origin cross-validation learns significantly better than non-personalized models (Figures 3 and 4). Our two-tiered deployment strategy offers a pragmatic solution to this personalization challenge, addressing the cold-start problem while incentivizing user engagement to unlock more accurate, individualized predictions.

Second, our expert evaluation suggests that web-retrieval is likely a critical component for high-quality health coaching. This is indicated in our finding of expert preferences for the web-retrieval agent (SePA-web) that, while at the margin of statistical significance (p=0.05), was practically meaningful (Cliff's $\delta$=0.30, medium effect). This highlights the value of grounding advice in external, verifiable knowledge while making the retrieval context-aware. By dynamically injecting ML-driven risk predictions into search queries, we ensure the retrieved evidence is directly relevant to the user's immediate physiological state, a key limitation we identified in prior systems like PHIA [5].

### B. Trust, Transparency, and Reproducibility

A central pillar of our work is addressing the black box problem prevalent in many commercial AI systems. While the full coaching system is part of an ongoing longitudinal study, we are committed to advancing reproducible research in this domain. By constraining retrieval to a curated domain whitelist, enforcing strict claim-level citation, and committing to release our Web-Retrieval pipeline implementation upon publication, we provide a transparent and reproducible blueprint for this critical system component. This contrasts with closed-source systems and offers a foundation upon which the community can build, scrutinize, and improve. The technical details and hyperparameters of this pipeline are detailed in Section III to facilitate replication.

### C. Practical Considerations and Performance Trade-offs

A practical consideration of our design is the trade-off between response quality and system latency. We quantified this by analyzing advice-seeking queries on our deployment server (16GB RAM CPU). As shown in Figure 5, enabling our web-retrieval pipeline increased the median response time from 4.41s (n=28) to 19.69s (n=135). This overhead is inherent to the multi-step retrieval process, which includes document fetching, processing with the `all-mpnet-base-v2` model, and reranking with the `ms-marco-electra-base` cross-encoder (see Section III). Our selection of these models represents a balance between retrieval quality and real-time interactivity within the constraints of typical web server hardware. While larger models could enhance retrieval accuracy, their latency would be prohibitive in a conversational context. This highlights a critical area for future optimization through techniques like model distillation to improve user experience without sacrificing guidance quality.

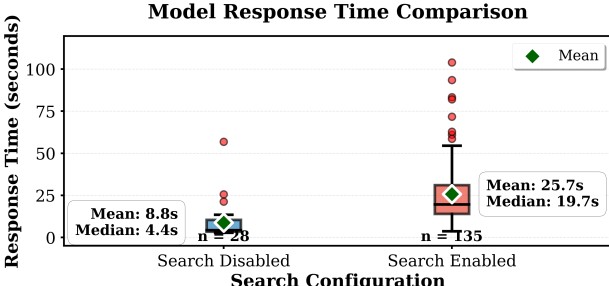

Fig. 5: System response time comparison. Box plots show response time distribution for agent turns with and without the web-retrieval pipeline for advice-seeking queries.

### D. Limitations and Future Work

This preliminary study has several limitations. Our predictive models were trained and validated on a cohort of collegiate student-athletes, and their generalizability to other populations requires further investigation. The expert evaluation, while providing valuable qualitative insights, was conducted with a small panel (n=4). Further research is needed to validate the system and predictive models on larger, more diverse cohorts. Furthermore, future work should look into enhancing the conversational capabilities of the coaching agent in light of the insights from our expert panel.

## VI. CONCLUSION

In this paper, we introduced SePA (Search-enhanced Predictive Agent), a novel LLM health agent built on the principles of transparency and evidence-based coaching to advance proactive health management. We demonstrated the critical need for personalization in predicting daily subjective states like stress, soreness, and injury risk, and presented a practical two-tiered modeling strategy to achieve this.

The core contribution of our work is a trustworthy, context-aware web-retrieval pipeline that leverages these daily predictions to find highly relevant and verifiable guidance from a curated set of expert sources. Our expert-driven evaluation demonstrated that integration of proactive prediction and trusted retrieval leads to a meaningful improvement in coaching quality. By providing a detailed architectural blueprint and committing to release the core web-retrieval component as open-source, we offer a reproducible foundation for health agent systems that are personalized, transparent, verifiable, and proactive in helping users manage their health.

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
