# OpenReview forum: "SePA: A Search-enhanced Predictive Agent for Personalized Health Coaching"
_IEEE.org/EMBS/BHI/2025/Conference — BHI 2025_

### Official Review · Reviewer_xMUd · 2025-06-26
**A promising and well-architected system for proactive health coaching whose claims of predictive performance require more rigorous methodological validation, specifically through robust cross-validation techniques.**

**Confidence:** 5
**Clarity Of Writing:** great
**Clinical Significance:** good
**Methodological Novelty:** fair
**Overall Rating:** 5
**Final Rating:** 7

**Experiments And Results:**

fair

**Questions For The Authors:**

1. Implement Robust Time-Series Cross-Validation: This is the most critical modification needed. To validate the performance of the personalized models (Figure 3), the authors must replace the current single-split evaluation with a proper time-series cross-validation technique. Methods like rolling-origin validation (or rolling forecast origin) would be appropriate. This involves creating multiple train/test splits for each user's time-series data (e.g., train on days 1-15, test on day 16; train on days 1-16, test on day 17, etc.) and averaging the performance metrics across all splits. This will produce a much more reliable and defensible estimate of the models' true predictive power. Without this, the central claim about the personalized models' performance (R-squared > 0.40) is not adequately supported.

2. Strengthen the Discussion of Limitations: The authors need to more strongly and prominently acknowledge the limitations related to the small, homogeneous sample size for both the predictive modeling and the expert evaluation. The manuscript should explicitly state that the results are preliminary and that further research is required to validate the SePA system on larger, more diverse populations before its findings can be generalized.

3. The authors should acknowledge and discuss the implications of the low inter-rater reliability for several questions in Table I. This transparency is necessary to properly contextualize the results of the expert evaluation.

4. The discussion of the borderline p-value (p=0.05) should be more nuanced. While the effect size (Cliff's δ) is noted, the authors should be more cautious in their claims, stating that the result provides suggestive, rather than conclusive, evidence and warrants further investigation with a larger panel of experts.

5. The abstract and introduction could more clearly state the specific cross-validation methods used for both the personalized and generalized models to improve upfront clarity.

**Strengths:**

1. Novel System Architecture: The integration of proactive risk prediction with a context-aware RAG pipeline is a significant strength. By using daily ML predictions to enrich search queries, the system moves beyond generic retrieval to find evidence directly relevant to the user's predicted physiological state.

2. Pragmatic Two-Tiered Model: The proposed strategy of deploying a generalized model for cold-start scenarios and unlocking a more accurate personalized model with user engagement is a practical and well-reasoned solution to the personalization challenge in real-world applications.

3. Emphasis on Trust and Transparency: The use of a curated whitelist of trusted domains, the enforcement of in-line citations, and the commitment to open-sourcing the retrieval pipeline are commendable design choices that directly address the critical need for trust and reproducibility in health AI.

4. Expert-Driven Evaluation: The inclusion of a blind evaluation with domain experts provides strong qualitative and directional quantitative evidence (Cliff's δ=0.3) for the value of the web-retrieval pipeline, adding a crucial layer of validation beyond automated metrics.

**Summary Of The Paper:**

This paper introduces SePA, a novel health coaching agent that integrates two key technologies: (1) personalized machine learning models to proactively forecast daily stress, soreness, and injury risk from wearable data, and (2) a retrieval-augmented generation (RAG) pipeline that grounds the agent's advice in a curated whitelist of trusted web domains. The authors propose a two-tiered system, starting with a general model for new users and transitioning to a higher-performing personalized model after 15 days of data collection. The evaluation consists of two parts: a quantitative comparison of the predictive models and a qualitative study with four domain experts who rated the RAG-enhanced advice as meaningfully superior to a non-retrieval baseline. The paper presents a transparent architecture and discusses practical trade-offs, positioning SePA as a blueprint for trustworthy AI in personal health.

**Weaknesses:**

1. Lack of Robust Cross-Validation: The primary analysis of personalized model performance (Figure 3) relies on a single train-test split for each participant (training on the first d days, testing on the remaining 45-d days). This method is susceptible to the specific data split and does not provide a reliable estimate of generalization performance over time. The results could be artificially inflated or deflated due to this single, static split.

2. Small and Homogeneous Sample Size: The study's conclusions are drawn from a small cohort of 28 collegiate student-athletes. This limits the generalizability of the findings to other populations (e.g., older adults, sedentary individuals, or those with chronic conditions). The expert panel for the qualitative evaluation is also very small (n=4).

3. Borderline Statistical Significance: The p-value for the expert evaluation, which is a cornerstone of the paper's contribution, is p=0.05. This is at the very margin of statistical significance and could be interpreted as weak evidence. A larger expert panel might be needed to confirm this finding more decisively.

4. Low Inter-Rater Reliability for Some Questions: In Table I, several of the questions used for the expert evaluation have very low or zero inter-rater reliability (Kendall's W of 0.00 or 0.25). This indicates a lack of consensus among the experts on how to evaluate the answers to these questions, which undermines the validity of the results for those specific items.

---

### Official Review · Reviewer_kg8i · 2025-07-15
**This paper presents a promising approach to personalized health coaching offering personalized guidance grounded in contextual evidence.**

**Confidence:** 3
**Clarity Of Writing:** great
**Clinical Significance:** good
**Methodological Novelty:** good
**Overall Rating:** 6
**Final Rating:** 6

**Experiments And Results:**

great

**Questions For The Authors:**

1. Did you refer Fig. 2 anywhere?
2. How does the system handle potentially conflicting or outdated information retrieved from the curated source whitelist, especially in a rapidly evolving health domain?
3. Did your implementation leverage any open-source LLMs for generation or retrieval?

**Strengths:**

1. The paper demonstrates that personalized predictive models, especially the PHM model, significantly outperform generalized baselines in forecasting wellness risks like stress, soreness, and injury, which highlight the effectiveness of individualized modeling in health applications.
2. By integrating retrieval-augmented LLMs with expert validations, the system offers trustworthy, context-relevant coaching that addresses a key limitation in existing black-box wellness tools.
3. Fig. 1 is quite comprehensive.

**Summary Of The Paper:**

This paper introduces SePA, a personalized health coaching system that combines predictive modeling with retrieval-augmented LLMs to provide context-aware, evidence-based guidance. The evaluation shows that personalized models, particularly with sufficient user history, significantly outperform generalized baselines in predicting stress, soreness, and injury risk, validating the system’s two-tiered deployment strategy. While the core contributions are relevant and well-motivated, the paper would benefit from broader evaluation (along with the expert validation) and clearer presentation to strengthen its overall impact.

**Weaknesses:**

1. The predictive models were trained and validated on a relatively small sample of participants (n=28, Fig. 3), limiting the generalizability of the findings to broader or more diverse populations.
2. Similarly the expert-driven evaluation of the retrieval-augmented coaching was conducted with only four experts, which may not be sufficient to draw robust conclusions about the system’s practical usability across varied real-world settings.
3. The paper does not provide comparisons with existing commercial/academic health coaching systems, which could help better assess SePA’s performance and contributions.
4. Some more description on Fig. 4 could be beneficial.

---

### Official Review · Reviewer_cFJj · 2025-07-16
**A novel system that integrates predictive modeling with retrieval-augmented generation (RAG) and large language models (LLMs) to deliver trustworthy, personalized, and proactive health coaching based on wearable sensor data.**

**Confidence:** 4
**Clarity Of Writing:** poor
**Clinical Significance:** excellent
**Methodological Novelty:** great
**Overall Rating:** 7

**Experiments And Results:**

great

**Questions For The Authors:**

1. Although the retrieval pipeline is restricted to trusted sources (e.g., PubMed, institutional websites), health recommendations can still vary. How does the system handle conflicting advice across sources?
2. Are the sources used in the final response explicitly shown to users, allowing them to verify and trace the origin of the recommendations?
3. Do users receive any form of transparency or confidence metric (e.g., model certainty or prediction reliability) alongside risk scores or generated advice?

**Strengths:**

1. Shift from retrospective analytics to daily forecasting enables predictiable and actionable health coaching.
2. The trusted web-retrieval pipeline ensures high factual accuracy and verifiability, addressing common concerns with LLM hallucinations. Also enable user to read and understand the complicated data.
3. Combines quantitative performance evaluation with expert ratings and qualitative feedback, showing both statistical and practical value.

**Summary Of The Paper:**

This paper introduces SePA, a predictive AI agent that combines individualized forecasting with retrieval-augmented generation (RAG) to deliver personalized and proactive health coaching. The system first predicts daily stress, soreness, and injury risk using wearable sensor data from 28 participants, with personalized models (activated after 10–15 days of user input) significantly outperforming generalized models (R² > 0.40 for stress and injury risk). Building on these predictions, SePA integrates a context-aware web retrieval pipeline that dynamically rewrites queries based on user-specific context and grounds LLM responses in expert-vetted sources. In a blind evaluation, domain experts preferred SePA’s retrieval-augmented advice over non-retrieval outputs, highlighting its relevance, trustworthiness, and practical utility. These results demonstrate SePA’s potential to advance personalized digital health coaching.

**Weaknesses:**

1. The dataset consists exclusively of collegiate student-athletes, which may limit the generalizability of the findings to broader populations, such as older adults or individuals with chronic health conditions.
2. The expert evaluation involved only four reviewers. While their feedback is valuable, the small sample size constrains the statistical strength and broader applicability of the conclusions.
3. The qualitative feedback from domain experts (Table 3) reflects general user impressions rather than targeted evaluations of response accuracy. For instance, in queries such as Q2 (“What nutritional strategies might help improve VO2?”), the availability of well-established references should allow for objective assessment of the system’s recommendations. Experts could have been instructed to evaluate factual correctness against known literature rather than providing usability-oriented comments.

---

### Official Review · Reviewer_aMjX · 2025-07-17
**SePA Shows Potential, but Key Claims Lack Full Support**

**Confidence:** 3
**Clarity Of Writing:** fair
**Clinical Significance:** fair
**Methodological Novelty:** good
**Overall Rating:** 3
**Final Rating:** 5

**Experiments And Results:**

poor

**Questions For The Authors:**

In Figure 3, the x-axis (number of training days) only goes up to 30, despite the dataset containing 45 days of data per participant. It is unclear why the model comparison results beyond day 30 are not shown. This omission is particularly concerning in Figure 3(a), where the RF model’s performance appears to approach or even match that of the proposed method toward the end. This raises a reasonable concern that results beyond day 30 may not support the paper’s core claim—that the personalized model consistently outperforms baselines in predicting injury risk, stress, or soreness. The paper should clarify whether this truncation was due to data limitations or selective reporting, and ideally include full 45-day results to allow a fair and complete assessment. (Scores could obviously rise if the doubt is clarified.)

Does the LLM-based health coaching pipeline utilize the predicted results for stress, soreness, and injury risk? Please explain the relationship between the wearable-based model prediction and the retrieval pipeline more clearly. (May not change the score.)

**Strengths:**

The system design is technically thorough, with clear architectural details and reproducibility plans.

The pipeline for enhancing the trust of the LLM-based health coaching agent by combining personalized proactive health states with constrained professional resources for RAG could be expanded to other application fields in connected health.

Expert feedback provides initial validation of practical usefulness, and the commitment to open-source release is commendable.

**Summary Of The Paper:**

The paper studies the challenge of transforming wearable health data into proactive, personalized, and trustworthy health coaching. It proposes SePA, an agent that combines predictive models for stress, soreness, and injury risk with an LLM-based health coaching platform. The predictive component uses a two-tier strategy—generalized models for new users and personalized models once sufficient user history is available. The coaching pipeline employs retrieval-augmented generation with query rewriting based on personalized physiological predictions to deliver more reliable suggestions. Results show that personalized models outperform general ones, and expert evaluations favor retrieval-based responses over non-retrieval outputs.

**Weaknesses:**

The figures—especially Figure 1—are difficult to read due to small text and too much information packed into a single image. The main message of the figure is unclear, and the section related to the coaching platform does not appear to convey any meaningful or relevant details. The figure would be easier to understand if it were split into at least two separate visuals: one showing the overall three-part system structure, and another focusing specifically on the Web Retrieval Pipeline. The text is also too small to read in other figures.

The paper relies on a narrow dataset of 28 athletes, limiting generalizability beyond this population. While expert ratings favor retrieval-based outputs, the small evaluation sample (n=4) and borderline statistical significance (p=0.05) reduce confidence in robust conclusions. But it's still okay as a pilot study for a conference paper, and the user has claimed this weakness in the discussion.

There are some typos or unclear writing. For example, in section B-2, should "This neural network architecture 2" refer to Fig.2? What are the participant-specific embeddings? There is no introduction of the contents or format of the term before; it just shows up without any explanation.

The paper length exceeds the 6-page limitation.

---

### Official Review · Reviewer_zu7m · 2025-07-20
**Effective utilization of LLM agent**

**Confidence:** 3
**Clarity Of Writing:** good
**Clinical Significance:** great
**Methodological Novelty:** good
**Overall Rating:** 5
**Final Rating:** 6

**Experiments And Results:**

good

**Questions For The Authors:**

Please see the limitations.

**Strengths:**

1. The system is evaluated both quantitatively and qualitatively.
2. Utilizes personalized time-series and self-reported data.
3. The LLM is utilized effectively.

**Summary Of The Paper:**

This paper introduces SePA, which is an LLM health coaching system for individuals.

**Weaknesses:**

1. How different features are processed and integrated is not very clear.
2. How privacy is going to be ensured when using external APIs and LLMs should be discussed more thoroughly.
3. This paper lacks examples produced by the LLMs and their corresponding qualitative analysis.